# Comparative Studies of Palmatine with Metformin and Glimepiride on the Modulation of Insulin Dependent Signaling Pathway In Vitro, In Vivo & Ex Vivo

**DOI:** 10.3390/ph15111317

**Published:** 2022-10-25

**Authors:** Okechukwu Patrick Nwabueze, Mridula Sharma, Abbirami Balachandran, Anand Gaurav, Anis Najwa Abdul Rani, Jeleń Małgorzata, Morak-Młodawska Beata, Charlie A. Lavilla, Merell P. Billacura

**Affiliations:** 1Department of Biotechnology, Faculty of Applied Sciences, UCSI University, Federal Territory of Kuala Lumpur 56000, Malaysia; 2Faculty of Pharmaceutical Sciences, UCSI University, Federal Territory of Kuala Lumpur 56000, Malaysia; 3Faculty of Pharmaceutical Sciences, Department of Organic Chemistry, Medical University of Silesia, Jagiellonska Str. 4, 41-200 Sosnowiec, Poland; 4Chemistry Department, College of Science & Mathematics, Mindanao State University-Iligan Institute of Technology, Iligan City 9200, Philippines; 5Department of Chemistry, College of Natural Sciences and Mathematics, Mindanao State University-Main Campus, Marawi City 9700, Philippines

**Keywords:** T2DM, palmatine, metformin, glimepiride, IRS1, PI3K, AKT2, GLUT4

## Abstract

(1) Insulin resistance, a symptom of type 2 diabetes mellitus (T2DM), is caused by the inactivation of the insulin signaling pathway, which includes IRS-PI3K-IRS-1-PKC-AKT2 and GLUT4. Metformin (biguanide) and glimepiride (sulfonylurea) are both drugs that are derivatives of urea, and they are widely used as first-line drugs for the treatment of type 2 diabetes mellitus. Palmatine has been previously reported to possess antidiabetic and antioxidant properties. (2) The current study compared palmatine to metformin and glimepiride in a type 2 diabetes model for ADME and insulin resistance via the PI3K/Akt/GLUT4 signaling pathway: in vitro, in vivo, ex vivo, and in silico molecular docking. (3) Methods: Differentiated L6 skeletal muscle cells and soleus muscle tissue were incubated in standard tissue culture media supplemented with high insulin and high glucose as a cellular model of insulin resistance, whilst streptozotocin (STZ)-induced Sprague Dawley rats were used as the diabetic model. The cells/tissue/animals were treated with palmatine, while glimepiride and metformin were used as standard drugs. The differential gene expression of PI3K, IRS-1, PKC-α, AKT2, and GLUT4 was evaluated using qPCR. (4) Results: The results revealed that the genes IRS-PI3K-IRS-1-PKC-AKT2 were significantly down-regulated, whilst PKC-α was upregulated significantly in both insulin-resistant cells and tissue animals. Interestingly, palmatine-treated cells/tissue/animals were able to reverse these effects. (5) Conclusions: Palmatine appears to have rejuvenated the impaired insulin signaling pathway through upregulation of the gene expression of IRS-1, PI3K, AKT2, and GLUT4 and downregulation of PKC-expression, according to in vitro, in vivo, and ex vivo studies.

## 1. Introduction

Type 2 diabetes mellitus (T2DM) is a heterogeneous metabolic disease characterized by impaired pancreatic beta cell insulin secretion, increased hepatic glucose uptake, and insulin resistance in peripheral tissues such as the liver, skeletal muscle, and adipose tissue [1,2]. According to the International Diabetes Federation, the global population of people with diabetes has risen from 451 million in 2017 to 463 million in 2019 and is estimated to rise to 700 million by the year 2045 [3]. Through the activation of the tyrosine kinase receptor pathway [4,5], insulin plays an important role in the regulation of plasma glucose levels as well as in promoting cellular glucose uptake, glycogenesis, and protein synthesis in skeletal muscle tissues. Impairment of glucose transport in diabetic skeletal muscle tissues induces insulin resistance and can lead to many serious complications, such as hyperglycemia and hyperlipidemia. Two distinct signaling pathways are responsible for the translocation of glucose transporter type 4 (GLUT4) to the cell membrane and, therefore, glucose uptake into the skeletal muscle tissue: (a) PI3K/AKT (phosphatidylinositol 3-kinase/protein kinase B) and (b) AMPK (AMP-activated protein kinase) pathways [6,7]. Insulin facilitates glucose uptake by increasing the translocation of GLUT4 from an intracellular pool to the plasma membrane through the activation of the PI3K/AKT pathway [8]. Derivatives of urea, such as metformin (biguanide) and glimepiride (sulfonylurea), have been used as first-line drugs for the treatment of T2DM. At the cellular level, it has been reported that sulfonylureas and metformin enhance insulin-mediated glucose utilization in muscle tissue by a distal mechanism to the insulin receptor.

Palmatine, a plant alkaloid extracted from the stem of *Coscinium fenestratum*, has been previously reported to possess in vitro and in vivo antidiabetic and antioxidant properties as well as offer protection of the kidney and liver [9,10,11,12]. Palmatine has also been shown to inhibit the in vitro and in vivo formation of advanced glycation end products (AGEs). Okechukwu et al. (2021) discovered that palmatine inhibited the upregulation of the GRP78 (glucose regulatory protein 78) and CALR (calreticulin) proteins in STZ-induced diabetic rats [13]. However, the molecular mechanism behind the antidiabetic action of palmatine is not fully understood. Therefore, the current study aimed to evaluate the molecular action of palmatine on the insulin signaling pathway in L6 skeletal muscle cells (in vitro), STZ-induced diabetic rats (in vivo), and soleus muscle tissue (ex vivo) [14].

The present study compared palmatine to metformin and glimepiride in a type 2 diabetes model on ADME analysis and insulin resistance via the PI3K/AKT/GLUT4 signaling pathway: in vitro, in vivo, ex vivo, and in silico molecular docking. The insulin-resistant cellular model involved the treatment of L6 skeletal muscle with high insulin and high glucose, whilst streptozotocin (STZ)-induced Sprague Dawley rats were used as the diabetic model. Glimepiride and metformin were used as standard drugs. In this study, the differential gene expression was evaluated using quantitative polymerase chain reaction (qPCR). In silico molecular docking was performed using Autodock Vina 1.2.0. The molecular descriptor, ADME parameters of Lipinski, Ghose, and Veber rules, and molecular target were calculated using the Swiss ADME server and Swiss Target Prediction server. The L6 skeletal muscle cell line was preferred because the targeted genes PI3K, AKT, and GLUT4 are well expressed in skeletal muscle cells. Molecular docking was used to model the interaction between palmatine and PI3K alpha and gamma at the atomic level and compare the same with the standards. This allows us to characterize the behavior of palmatine in the binding site of target proteins and compare the same with standards and shed light on the fundamental biochemical process behind the binding of ligand and target protein. The docking studies also provide additional evidence about the target proteins, i.e., PI3K alpha and gamma.

## 2. Results

The percentage relative gene expressions of IRS1, PI3K, PKC-α, AKT2, and GLUT4 of the diabetic control group and all drug-treated groups were plotted in Figure 1, Figure 2 and Figure 3. The effect of palmatine against the standard drugs (metformin and glimepiride) and the diabetic control group on each gene has been explained individually in Section 2.1, Section 2.2, Section 2.3, Section 2.4 and Section 2.5.

### 2.1. Effect of Palmatine on the Expression of IRS1

The upregulation of IRS1 observed in the drug-treated groups in all three models was higher than in the diabetic control group, with a percentage-fold increase of 85% to 90% (*p* < 0.0001). Among the treated groups, palmatine showed no significant difference with metformin in the cell culture and ex vivo models, but there was a 33%-fold difference between metformin and palmatine groups in the in vivo model. On the other hand, glimepiride showed a 29% and 33%-fold increase against palmatine in the cell culture and in vivo models, while palmatine showed a 40%-fold-increase against glimepiride in the ex vivo model.

### 2.2. Effect of Palmatine on the Expression of PI3K

The upregulation of PI3K in the palmatine-treated group of the cell culture model was the highest among all groups and was significantly different (*p* < 0.0001) from the diabetic control group (89%), glimepiride-treated group (81%), and metformin-treated group (80%). In the in vivo model, the palmatine-treated group showed a 66% increase against the diabetic control group, while glimepiride and metformin showed higher PI3K expressions (78% and 77%-fold increase, respectively) against the diabetic control group. Lastly, the ex vivo model showed that palmatine increased PI3K expression by 30% while glimepiride and metformin were able to express higher PI3K levels against the diabetic control group (39% and 41%, respectively).

### 2.3. Effect of Palmatine on the Expression of PKC-α

The expression of PKC-α in all three models was found to be downregulated in all treated groups as compared to the diabetic control group (*p* < 0.0001). Among the treated groups, there was no significant difference in all three drug-treated groups of the cell culture model and between the metformin and palmatine-treated groups of the ex vivo model. It was observed that the PKC-α expression in the in vivo model was significantly lower in the palmatine-treated group as compared to the glimepiride and metformin-treated groups (*p* < 0.0001). Among all three models, palmatine showed the highest PKC- downregulation in the cell culture model.

### 2.4. Effect of Palmatine on the Expression of AKT2

The palmatine-treated group showed a significant difference (*p* < 0.0001) in all models against the diabetic control group. Among the drug-treated groups, metformin showed the highest expression of AKT2 in all three models, followed by the palmatine-treated group in both cell culture and in vivo models with a difference of 62% and 53%, respectively. However, the palmatine-treated group was not significantly different from the glimepiride-treated group in the ex vivo model. Among all three models, palmatine showed the highest effect in expressing AKT2 in the in vivo model.

### 2.5. Effect of Palmatine on the Expression of GLUT4

GLUT4 was greatly expressed in all drug-treated groups of the three models as compared to the diabetic control group. Metformin expressed the highest GLUT4 levels, followed by palmatine and, lastly, glimepiride. In the cell culture model, the GLUT4 expression in the palmatine-treated group was 12% lower than that of the metformin-treated group, but it was 36% higher than the glimepiride-treated group. Similarly, there was a 15% reduction of GLUT4 expression in the palmatine-treated group against the metformin-treated group in the in vivo model. However, the GLUT4 expression of the palmatine-treated group was like that of the glimepiride-treated group by a trace difference of 1.1%. Lastly, the palmatine-treated group showed a lower GLUT4 expression against the metformin-treated group by 10.6%, while it was approximately 7% higher than that of the glimepiride-treated group.

### 2.6. In Silico Results

The RMSD values for the co-crystallized ligand pose vs. docked pose are documented in Table 1, while the overlaid poses are shown in Figure 4. The RMSD value lower than 2.0 Å was considered a validation of the docking procedure [15,16]. Acceptable RMSD values were obtained for the docking protocol adopted, indicating the ability of the docking protocol to reproduce the original co-crystalized pose [17,18]. The binding energies obtained via the docking of the ligands with PI3K alpha and PI3K gamma using AutoDock Vina 1.2.0 are shown in Table 2.

Based on the binding energies, it can be concluded that glimepiride, as the standard, exhibits the lowest binding energy for both receptors, i.e., −10 kcal/mol for PI3K gamma and −8.3 kcal/mol for PI3K alpha. Palmatine showed binding energies of −8.2 kcal/mol for PI3K gamma and −9.2 kcal/mol for PI3K alpha, which is lower than that of metformin, i.e., 5.0 kcal/mol for PI3K gamma and −4.9 kcal/mol for PI3K alpha. Thus, it can be said that palmatine has a good affinity for both PI3K alpha and PI3K gamma, which is higher than metformin and quite like glimepiride. Both standards used showed higher selectivity towards PI3K gamma compared to palmatine, which showed higher selectivity towards PI3K alpha.

Further studies were conducted on the binding interactions of glimepiride and metformin with PI3K gamma and alpha as shown in Figure 5. These complexes were viewed using Biovia Discovery Studio Visualizer 2021. Analysis of the binding interactions showed that glimepiride formed conventional hydrogen bonds with residues HIS 295, ASN 299, and ARG 849 of PI3K gamma while only with GLN 859 of PI3K alpha. This explains why binding energy is lower for PI3K gamma. There is also a pi-sigma interaction between glimepiride and TYR836 in PI3K alpha, while this was not the case for PI3K gamma. Hydrophobic interactions were observed between glimepiride and LEU 657, ARG 294, LYS 298, LEU 657, LEU 660, PHE 694, PHE 698 of PI3K gamma, and VAL 850, MET 922, ILE 932, PRO 778, ILE 800, LYS 802, ILE 848, and ILE 932 of PI3K alpha. The analysis of binding interactions of palmatine showed two conventional hydrogen bonds with SER 854 and ASP 933 of PI3K alpha, while one carbon-hydrogen bond with ALA885 of PI3K gamma.

There is also a Pi-Pi T-shaped interaction with aromatic rings of TYR780 and TYR386 of PI3K alpha and a Pi-sulphur interaction with MET 804 of PI3K gamma. Palmatine also forms Pi-sigma interactions with the residues MET 922 and ILE 932 of PI3K alpha, while this interaction was not seen with PI3K gamma. Palmatine showed hydrophobic interactions with both proteins, i.e., with LEU 838, CYS 869, ILE 879, LYS 833, ILE 879, MET 953, MET 804, TRP 812, TYR 867, ILE 879, ILE 963 of PI3K gamma, and with ILE 932, VAL 850, TRP 780, Ile 800, ILE 848 of PI3K alpha.

### 2.7. ADME Analysis

The tested compound, palmatine (**1**), was subjected to preliminary in silico analyses of pharmacokinetic parameters and ADMET profile and prediction of biological targets using the web platforms SwissADME and SwissTargetPrediction (Table 3). Metformin (**2**) and glimepiride (**3**) were used as reference compounds.

Due to the structural differences (Figure 6), all molecular descriptors from the molecular weight up to the TPSA surface and lipophilicity are different.

However, despite such fundamental differences, all molecules comply with Lipinski’s five rules, i.e., they show good oral bioavailability. The rules of bioavailability of Ghose, Veber, and Muegge are met only for palmatine (**1**), while compounds (**2**) and (**3**) do not fully meet these conditions.

Additionally, the program SwissTargetPrediction indicated for compounds a high probability of interaction with various molecular targets like hydrolase, kinase, cytochrome P450, and family A G protein-coupled receptors (Figure 7).

## 3. Discussion

T2DM is distinguished by three major defects: abnormal pancreatic insulin secretion, increased hepatic glucose uptake, and peripheral insulin resistance [2]. At a molecular level, insulin resistance is associated with the impairment of the glucose transport system, which shows a reduction of the intracellular pool of transporters [6,19]. The molecular signaling pathway of insulin consists of a series of activations of genes that eventually lead to the stimulation of GLUT4. It may be an insulin-dependent or insulin-independent pathway. The insulin-dependent pathway involves the proteins IRS-PI3K- PKC-α -AKT-GLUT4, while the insulin-independent pathway utilizes the protein AMPK [20,21].

The impairment of these insulin signaling pathways is the major cause of T2DM. Activation of inactivation of these major proteins through phosphorylation or dephosphorylation plays a role in aggravating insulin signaling abnormalities that lead to insulin resistance [22]. Hence, compounds that increase the peripheral sensitivity to insulin are useful in the treatment of T2DM. The inhibition of the PI3K/AKT insulin signaling pathway is caused by the upregulation of serine/threonine protein kinases such as the PKC-α and PKC-α isoforms [23,24].

This study evaluated the insulin-dependent signaling pathway with a focus on the IRS1, PI3K, PKC-α, AKT2, and GLUT4 genes. The skeletal muscle is the largest (by mass) organ of the human body and is the primary site of glucose uptake, disposal, and storage, accounting for approximately 75% of the entire body’s glucose uptake under insulin stimulation [25]. As a result, maintaining normal glucose levels requires proper skeletal tissue function [26,27]. The L6 skeletal muscle cell line and skeletal muscle tissue from STZ-induced diabetic rats were used in this investigation because they express a high level of GLUT4, which is required for glucose absorption from the plasma. The diabetic control group of all three models showed a decrease in the expression of IRS1, PI3K, AKT2, and GLUT4 and an increase in PKC-α.

Insulin expresses its action by binding with the insulin receptor (IR), which leads to the phosphorylation of IRS1. IRS proteins are the key center receptors in the insulin-dependent and insulin-independent pathways [28]. Physiologically, IRS is phosphorylated when insulin combines with its receptor on the surface of the skeletal muscle cell, followed by PI3K activation. The activated PI3K can catalyze PIP2 (phosphatidylinositol 4,5-biphosphate) to PIP3 (phosphatidylinositol (3,4,5)-trisphosphate), which then activates the downstream signaling factor AKT. Phosphorylated AKT translocates GLUT4 to the plasma membrane to promote glucose uptake into the cells, which contributes to decreasing plasma glucose concentrations [29]. These phosphoinositides are responsible for the activation of AKT via PDK1 in the skeletal muscle tissues for glucose uptake [29,30]. Any disruptions in the PI3K signal transduction pathway may result in IR [31]. PI3K is an intracellular lipid kinase and an element of the cell membrane.

Hypoactivation or impairment of IRS will hamper the activation of PI3K/AKT-GLUT4, which may result in IR with T2DM. Świderska et al. (2018) report that there is usually a decrease in the expression of IRS1 in T2DM, which causes a malfunction in the secretion of insulin [24]. Many studies have linked hypoactivation or malfunction of IRS-PI3K-AKT-GLUT4 and upregulation of PKC- to increased oxidative stress, free fatty acids, and lipid levels, which may lead to beta cell destruction and alter insulin release [32]. When compared to the diabetic control group, palmatine reversed the expression of IRS-PI3K-AKT-GLUT4 and PKC- in high glucose and high insulin in all three models, indicating rejuvenation of the impaired insulin signaling pathway.

Palmatine has been previously reported to possess in vitro and in vivo antioxidant, antidiabetic, and antiglycemic properties [9,10,12,33]. The upregulation of the IRS1 gene by palmatine could be due to these properties [24,32]. Antioxidants interact with reactive free radicals such as oxygen and nitrogen species to neutralize their oxidative damaging effects. Neutralization by antioxidants involves the destruction of the oxidative cascade chain reactions in the cell membranes to fine-tune the level of free radicals. Many studies have shown that antioxidants do not only mitigate oxidative stress and improve stem cell survival but also affect the potency and differentiation of these cells, thus increasing genomic stability, restoration, and regeneration of cells through the balance between the generation and degradation of reactive oxygen species (ROS) within the cells and tissues [34,35].

Advanced glycation end product may contribute to the downregulation of IRS-PI3K-AKT-GLUT4 expression because they slowly and irreversibly form on proteins that are exposed to carbonyl and substrate stress, especially under conditions of hyperglycemia, hyperlipidemia, and/or oxidative stress [36]. AGEs play an important role in the pathophysiological processes affecting patients with diabetes, Alzheimer’s disease, and aging [37,38,39]. AGEs are known to cause adverse side effects on the growth of several cells, such as vascular endothelial cells and renal tubular epithelial cells [40,41,42,43,44,45,46,47,48]. Several studies have suggested that AGEs mediate cell apoptosis and may play an important role in the pathogenesis of biophysical disorders [36,49].

Elevated AGEs in diabetic patients may cause a number of pathological changes, including the promotion of endothelial progenitor cells (EPC) and endothelial cell apoptosis [47,50]. AGE-HSA stimulated transcription factors like NF-B, NF-AT, and AP-1. AGE acts through its cell surface receptor, RAGE, and degranulates vesicular contents, including interleukin-8 (IL-8). The number of RAGEs, as well as the amount of NF-κB activation, is low, but cell death is higher in neuronal cells upon AGE treatment. Degranulated IL-8 acts through its receptors, IL-8Rs, and induces sequential events in cells: an increase in intracellular Ca (2+), activation of calcineurin, dephosphorylation of cytoplasmic NF-AT, nuclear translocation of NF-AT, and expression of FasL. Expressed FasL increases caspase activity and induces cell death. Interaction between AGE and RAGE also increases the generation of ROS. Palmatine has been reported to scavenge free radicals in vitro and modulate in vivo antioxidant enzymes. Palmatine has also been reported to reduce lipids and triglycerides. In vitro and in vivo antioxidant, anti-hyperlipidemia, and antiglycation may have contributed to the reversal of the hypoactivation/malfunction of the IRS-PI3K-AKT-GLUT4 and PKC-α -insulin signaling pathways in high glucose and high insulin (in vitro and ex vivo) and STZ-induced diabetic rat model when compared to the insulin-resistant (diabetic) control group. 

Structural analyses showed that the tested compounds, although they exhibit antidiabetic activity, are very different. Palmatine is an isoquinoline derivative. Metformin is a urea derivative, specifically a biguanide, whereas glimepiride is a sulfonylurea derivative. The results of analyses of molecular descriptors and lipophilicity of these compounds are very divergent, which can be explained by their structural diversity. However, despite such fundamental differences, all molecules comply with Lipinski’s five rules. Additionally, the analyzed compounds showed a high probability of interaction with various molecular targets like hydrolase, kinase, cytochrome P450, and family A G protein-coupled receptors. The obtained preliminary in silico results confirmed the high probability of the compounds’ having an influence on biological targets. Below is the suggested schematic diagram of the mechanisms of action of palmatine in Figure 8.

## 4. Materials and Methods

### 4.1. Cell Culture

Cells of L6 rat skeletal muscle cells (ATCC-CRL-1458) procured from Bio-Focus Scientific, Malaysia, were maintained in DMEM media containing 10% (*v*/*v*) FBS and 1% penicillin (100 U/mL). The cells were cultured in a humidified atmosphere with 5% CO_2_ at 37 °C until they reached 70–80% confluency. Then they were detached using 0.25% (*w*/*v*) trypsin-EDTA and 0.05% glucose in PBS. The doubling time of the cells was 22–24 h. Therefore, re-suspended cells were transferred to a new T25 flask in a split ratio of 1:4 (subcultured in 2–3 days).

### 4.2. Cell Differentiation and Induction of Insulin Resistance

L6 rat skeletal cells were differentiated into myotubes by incubating the cells in DMEM media supplemented with 2% horse serum and 1% penicillin for seven days. All the cultures were grown in a T-25 flask in a humidified atmosphere with 5% CO_2_ at 37 °C. For the induction of insulin resistance, the method according to Zhou et al. (2016) was used [51]. In brief, differentiated myotubes were grown at a density of 2 × 10^4^ cells/well on 96-well microplates and were pretreated with the DMEM media containing 25 mM glucose and 100 mM insulin for 24 h. Cells were then switched to low-glucose (5 mM) DMEM without insulin for 5 h. Glucose-enriched media was discarded after 5 h and replaced with 100 mM insulin media for another 30 min. Glimepiride (2 μM), metformin (2 μM), and palmatine (2 μM) were prepared in the DMEM media with 2% of inactivated FBS. The cells were incubated with glimepiride (2 μM), metformin (2 μM), and palmatine (2 μM) for 24 h. After the treatment, RNA was extracted for the gene expression studies.

### 4.3. Animal Preparation

Healthy female adult Sprague-Dawley (SD) rats weighing 150–200 g and eight weeks old were used in this study. The rats were purchased from Sinar Scientific, Malaysia, and kept at the animal house of the Pharmacy Department, UCSI University, Malaysia, under standard conditions (at a temperature of 22 ± 1 °C; humidity 60–70%; 12 h dark/12 h light schedule), provided with free access to standard rat feed pellets and water. The animals were acclimatized to the laboratory conditions for two weeks before the commencement of the experiment. Experimental protocols were approved by the Universiti Kebangsaan Malaysia under the ethical code UCSI/2017/PATRICK/20-AUG./801-NOV-2018-DEC-2019.

### 4.4. Induction of Diabetes

Induced T2DM in rats was achieved with STZ (0.10 M, pH 4.5) in citrate buffer. An overnight fasted rat was given 50 mg/kg body weight (b.w.) of freshly prepared STZ solution intraperitoneally (i.p.). The plasma glucose levels were checked 72 h after the STZ injection to confirm that the rats were hyperglycemic. Rats with fasting blood glucose levels above 11.0 mM were used for the study [12].

### 4.5. Experimental Design (Grouping and Dosing of Animals)

A total of sixty SD rats were divided into ten groups, each group consisting of six rats. Group 1: Normal group-no induction of T2DM; rats were given saline; Group 2: Negative group-induced T2DM rats, treated with saline; Group 3: Positive control 1: -induced T2DM rats treated with 16 μM of metformin; Group 4: Positive control 2: -induced T2DM rats treated with 4.05 μM of glimepiride; Group 5: Test compound-induced T2DM treated with 5.85 μM of palmatine. All the treatments were orally administrated to the rats following the study design as above for a duration of 8 weeks. Plasma glucose levels and body weight were measured every week via withdrawal of blood from the aseptically cut tip of the tail using a glucometer and test strip, and a weighing balance was used to measure the weight of the animals [12].

### 4.6. Ex Vivo Model Preparation

Healthy female adult Sprague-Dawley (SD) rats weighing 150–200 g and aged eight weeks old were used in this study. The rats had a fasting blood glucose level of below 11.0 mmol/L and no physical defects. After two weeks of acclimatization in the laboratory facility, they were dissected, and soleus muscles (2 cm long) were extracted, snap-frozen, kept in a polypropylene vial, and stored at −80 °C for ex vivo study. Snap-frozen extracted soleus muscle tissue stored in a polypropylene vial at −80 °C was removed and allowed to thaw.

### 4.7. Induction of Insulin Resistance into Soleus Muscle Tissue

Isolated soleus muscle tissues were incubated in glass vials gassed (95% O_2_-5% CO_2_) containing 50 mL of KHB buffer supplemented with 0.1% bovine BSA and 1 mM pyruvate for 24 h at 37 °C. After the incubation, the KHB-BSA buffer was replaced by a KHB buffer containing 125 mmol/L glucose and 500 mmol/L insulin (High Glc/Ins) for 24 h. The buffer was replaced after 24 h with high glucose/insulin KHB-buffer, KHB-buffer containing 25 mmol/L of glucose for 5 h at 37 °C. After 5 h, the buffer was replaced with KHB-Glucose buffer containing 1000 mM insulin for 30 min.

Insulin-resistant induced soleus muscle tissues were treated with the drugs by incubating soleus muscle tissues with KHB-buffer containing the respective drugs according to the treatment regime. The soleus muscle tissues were ready for gene expression experiments and analysis after 24 h of incubation with the KHB-buffer containing the respective drugs [23].

### 4.8. Total RNA Isolation and Real-Time PCR

After incubation of L6 rat skeletal myotubes and insulin-resistant treated soleus muscle tissue with glimepiride, metformin, and palmatine for 24 h, the tissue was snap-frozen and stored. For the total RNA extraction, the frozen skeletal muscle tissues from each respective group were homogenized using liquid nitrogen. Total RNA was isolated using the innuPREP RNA mini kit (Analytik Jena, Jena, Germany) and Tris-sure extraction buffer (Bioline, Luckenwalde, Germany). The integrity and quantity of each extracted total RNA sample were measured by reading the absorbance of the samples at 260/280 nm using a NanoDrop 2000 spectrometer (Eppendorf, Hamburg, Germany). cDNA synthesis was done according to the manufacturer’s protocol of the SensiFast cDNA synthesis kit (Bioline, Camarillo, CA, USA). cDNA and primers were prepared with the SensiFast SYBR HI-ROX kit (Bioline, Camarillo, CA, USA) for real-time PCR. Expression of the following genes (details in Table 1) was studied: insulin receptor substrate-1 (IRS-1), phosphoinositide 3-kinase (PI3K), Protein Kinase C (PKC-α), Protein Kinase B (AKT2/PKB), and glucose transporter 4 (GLUT4). Gamma-actin (γ-actin) and beta-actin (β-actin) were used as housekeeping genes. A PCR mix (150 µL) was prepared for each gene and divided into six tubes containing a 25 µL reaction. The amplification conditions were 95 °C (2 min) for DNA polymerase activation, followed by 95 °C (5 s), 60 °C (10 s), and 72 °C (20 s). The annealing temperatures were 49, 51, 53, 55, 57, and 59 °C for 15 s for 40 cycles. The melt curve was done to see if there was unspecific binding or amplification. Β-actin and γ-actin were used as housekeeping genes, and they were stable. The data were analyzed using Applied Biosystems Step One Software Version 2.3.

### 4.9. In Silico Docking of Palmatine with PI3K

#### 4.9.1. Protein Preparation

The X-ray crystal structures of P13K alpha (PDB ID: 6PYS) and P13K gamma (PDB ID: 5JHB) were downloaded from the Protein Data Bank and prepared using a standard protocol. The structures were loaded into Biovia Discovery Studio 2021 Client. Heteroatoms such as water are removed. The binding site attributes were determined based on the location of the co-crystalized ligand. The co-crystallized ligands (P5J and 6K5) for each structure were removed, charges were then added, and the structures were then saved as prepared protein. The prepared structures were then loaded into AutoDock Tools version 1.5.7 [52] and saved in pdbqt format.

#### 4.9.2. Ligand Preparation

The structures of palmatine and two positive standards (glimepiride and metformin) were loaded into AutoDock Tools version 1.5.7 for preparation as ligands. Gasteiger charges were added to the respective structures and then saved in pdbqt format.

#### 4.9.3. Redocking of the Co-Crystallized Ligand

The co-crystallized ligand was prepared using the same method as the other ligands and then docked into the binding site of the respective proteins using the same method to obtain the docked poses. The docked poses were overlaid on the original poses in PyMol version 2.5.2 (Schrödinger) to calculate the Root Means Square Deviation (RMSD). An RMSD value lower than 2.0 Å was considered the validation of the docking protocol.

#### 4.9.4. Molecular Docking

Molecular docking was performed using Autodock Vina 1.2.0 [53]. The ligands prepared via the previous step were docked into the binding sites of the prepared P13K alpha and P13K gamma structures. The binding site coordinates were X: −18.018853, Y: 11.788912, Z: 28.460029, and X: 21.349173, Y: −4.074288, Z: 20.869615 for PI3K alpha and PI3K gamma, respectively. The binding site was covered using a grid of dimensions of 40 × 40 × 40 points. Exhaustiveness was set at 8. The pose with the lowest binding energy was identified as the most favorable pose. The binding interactions were then viewed using the Biovia Discovery Studio 2021 Client.

#### 4.9.5. ADME Method

The molecular descriptor, ADME parameters, parameters of Lipinski, Ghose, and Veber rules, and molecular target were calculated using the SwissADME server [http://www.swissadme.ch/index.php] (Access data: 18 April 2022) and the SwissTargetPrediction server [http://www.swisstargetprediction.ch/] (Access data: 18 April 2022).

### 4.10. Statistical Analysis

All data were expressed as mean ± SD using XL STAT version 14.0. The data were analyzed by one-way ANOVA with Duncan’s multiple-range test. Differences between groups were rated significant at a probability error (*p*) < 0.05.

## 5. Conclusions

The present findings, which were obtained using various cellular models and in silico molecular docking, provide compelling evidence for the role of palmatine in the prevention of T2DM and insulin resistance. Although both derivatives of urea, metformin (biguanide) and glimepiride (sulfonylurea), showed similar effects, the effect of palmatine was stronger compared to metformin and glimepiride. Palmatine’s potential mechanism of action is through reduced expression of PKC-α with a concurrent increase in the expression of tyrosine phosphorylation, activating the IRS-1-PI3K-AKT2 cascade and thus enhancing GLUT4 translocation. Based on the binding energies, it can be concluded that palmatine has a good affinity for both PI3K alpha and PI3K gamma compared to that of the derivatives of urea metformin and glimepiride. Preliminary in silico analyses of pharmacokinetic parameters and ADMET profile and prediction of biological targets using the web platform SwissADME and SwissTargetPrediction, revealed that palmatine showed very good oral bioavailability and obeyed the Ghose, Veber, and Muegge rules of bioavailability compared to metformin and glimepiride, which did not fully meet these conditions. Additionally, the program SwissTarget Prediction indicated a high probability of palmatine interaction with various molecular targets like hydrolase, kinase, cytochrome P450, and family A G protein-coupled receptors. The obtained preliminary in silico results confirmed the high probability of the compounds’ having an influence on biological targets.

## Figures and Tables

**Figure 1 pharmaceuticals-15-01317-f001:**
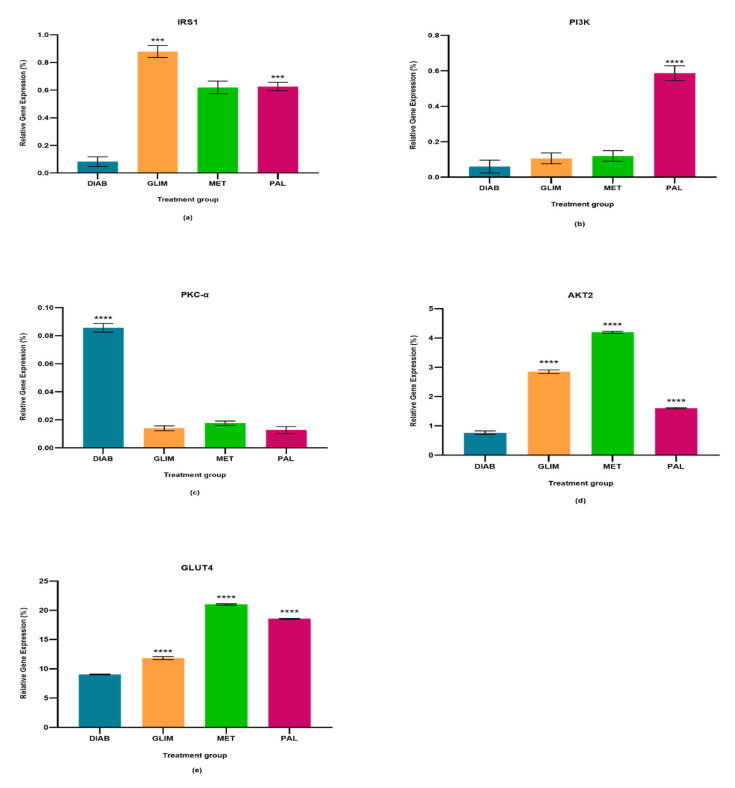
In the cell culture model, relative quantification of (**a**) IRS-1, (**b**) PI3K, (**c**) PKC-α, (**d**) AKT2, and (**e**) GLUT4 expressions in diabetic, glimepiride, metformin, and palmatine treatment groups. The gene was normalized against a geometric mean of two housekeeping genes (β-actin and γ-actin). All the data points were presented as mean SD. Palmatine group vs. other treatment groups: *** *p* < 0.001, **** *p* < 0.0001.

**Figure 2 pharmaceuticals-15-01317-f002:**
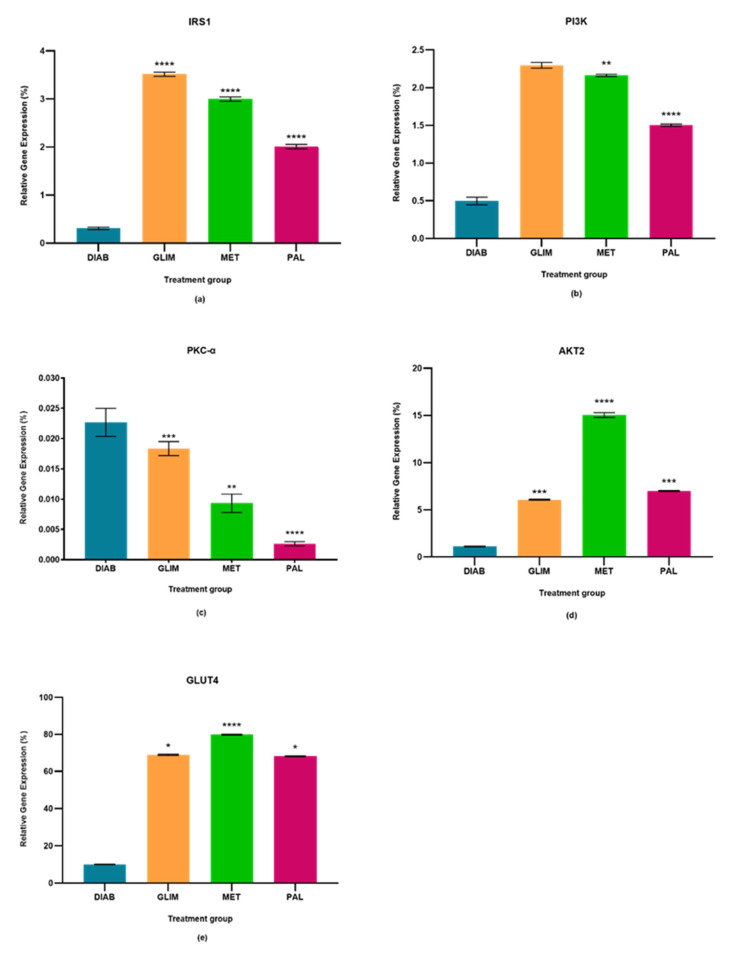
In the in vivo model, relative quantification of (**a**) IRS-1, (**b**) PI3K, (**c**) PKC-α, (**d**) AKT2, and (**e**) GLUT4 expressions in diabetic, glimepiride, metformin, and palmatine treatment groups. The gene was normalized against a geometric mean of two housekeeping genes (β-actin and γ-actin). All the data points were presented as mean ± SD. Palmatine group vs. other treatment groups: * *p* < 0.05, ** *p* < 0.01, *** *p* < 0.001, **** *p* < 0.0001.

**Figure 3 pharmaceuticals-15-01317-f003:**
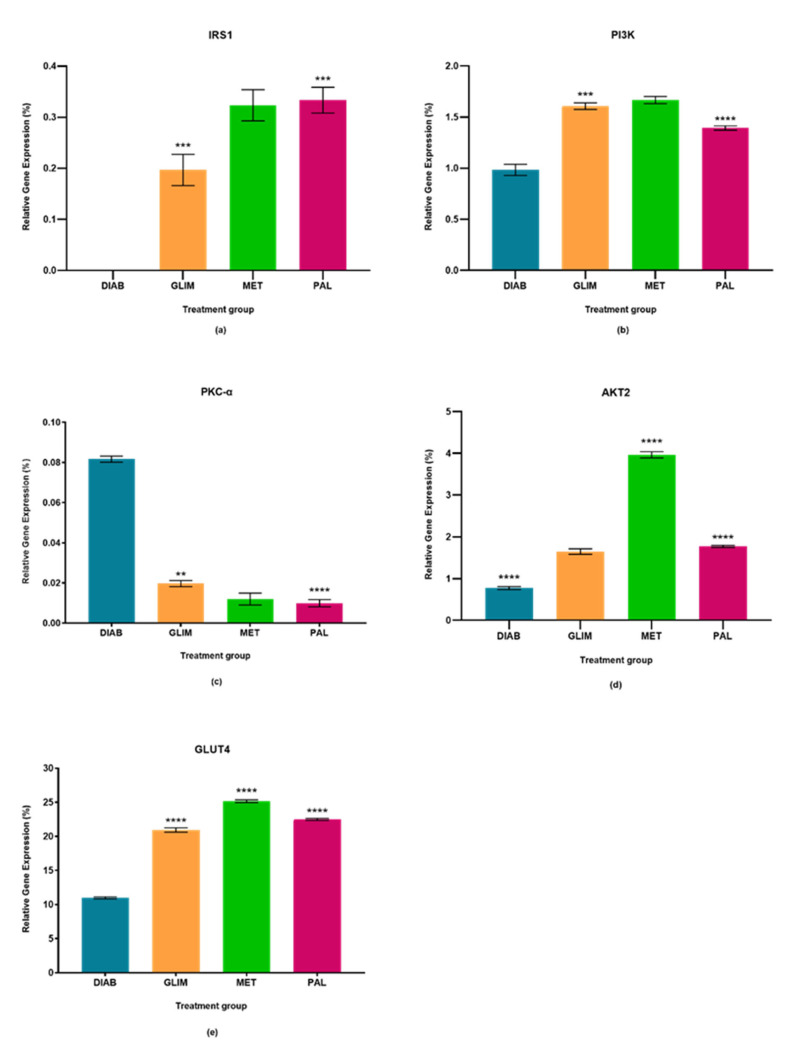
In the ex vivo model, relative quantification of (**a**) IRS-1, (**b**) PI3K, (**c**) PKC-α (**d**) AKT2, and (**e**) GLUT4 expressions in diabetic, glimepiride, metformin, and palmatine treatment groups. The gene was normalized against a geometric mean of two housekeeping genes (β-actin and γ-actin). All the data points were presented as mean ± SD. Palmatine groups vs. other treatment groups: ** *p* < 0.01, *** *p* < 0.001, **** *p* < 0.0001.

**Figure 4 pharmaceuticals-15-01317-f004:**
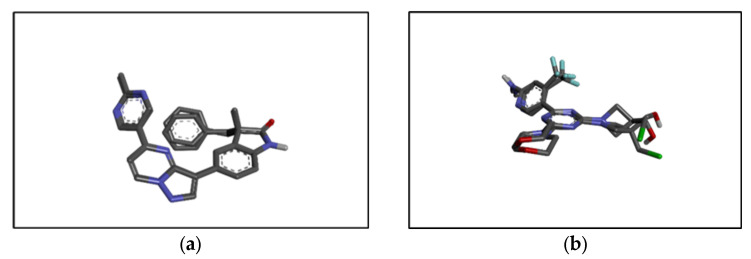
Structure of (**a**) PI3K alpha co-crystallized (from PI3K alpha) vs. docked pose and (**b**) PI3K gamma co-crystallized (from PI3K gamma) vs. docked pose.

**Figure 5 pharmaceuticals-15-01317-f005:**
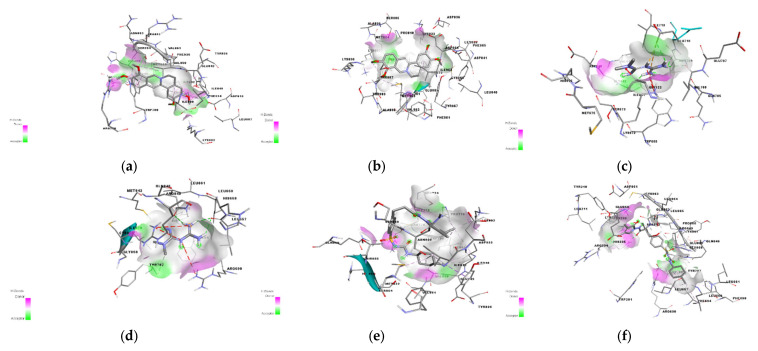
The structural binding interaction of (**a**) palmatine with PI3K alpha and (**b**) with PI3K gamma, (**c**) metformin with PI3K alpha and (**d**) PI3K gamma, and (**e**) glimepiride with PI3K alpha and (**f**) PI3K gamma.

**Figure 6 pharmaceuticals-15-01317-f006:**
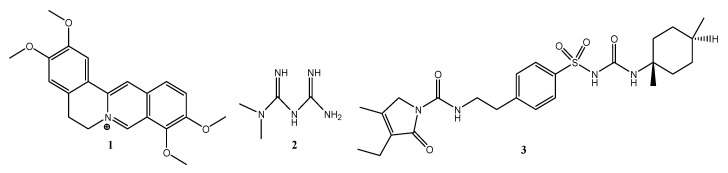
Chemical structures of palmatine (**1**), metformin (**2**) & glimepiride (**3**).

**Figure 7 pharmaceuticals-15-01317-f007:**
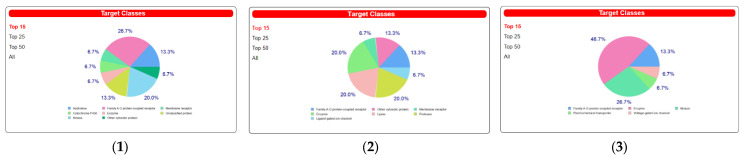
The target classes of palmatine (**1**), metformin (**2**), and glimepiride (**3**).

**Figure 8 pharmaceuticals-15-01317-f008:**
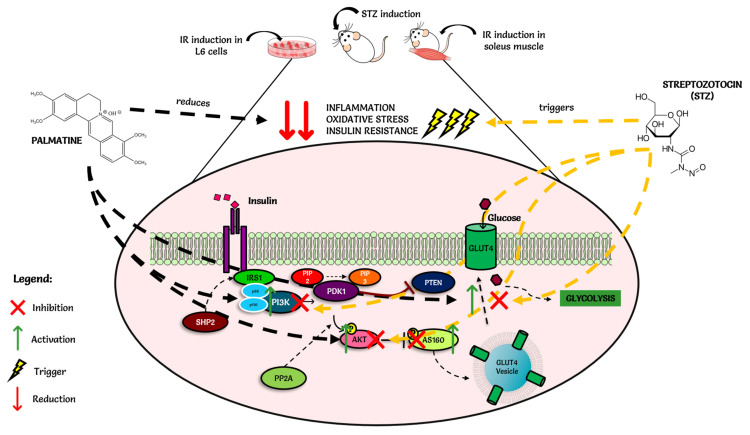
Schematic diagram of the mechanisms of action of palmatine.

**Table 1 pharmaceuticals-15-01317-t001:** RMSD value of co-crystallized (from PI3K Alpha) vs. docked poses.

Protein-Ligand	RMSD Value (Å)
PI3K alpha–P5J	0.292
P13K gamma–6K5	1.274

**Table 2 pharmaceuticals-15-01317-t002:** The binding energy of the drugs with PI3K Gamma and PI3K Alpha.

Item	Binding Energy (kcal/mol)	
	PI3K Gamma	PI3K Alpha
Glimepiride	−8.3	−10
Metformin	−5.0	−4.9
Palmatine	−8.2	−9.2

**Table 3 pharmaceuticals-15-01317-t003:** The molecular descriptor and parameters of Lipinski’s, Ghose’s, and Veber’s rules for palmatine (**1**), metformin (**2**) & glimepiride (**3**) data here.

No	Molecular Mass (M)	H-Bond Acceptors	H-Bond Donors	Rotatable Bonds	TPSA	LogP Consensus	Lipinski’s Rules	Ghose’s Rules	Veber’s Rules	Muegge’s Rules
**1**	352	4	0	4	40.80	2.64	+	+	+	+
**2**	129	3	2	4	36.93	0.75	+	-	+	-
**3**	490	11	5	3	133	2.76	+	-	-	+

## Data Availability

Data is contained within the article.

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
