# Peer review of "Comparative Studies of Palmatine with Metformin and Glimepiride on the Modulation of Insulin Dependent Signaling Pathway In Vitro, In Vivo & Ex Vivo"

_pharmaceuticals, 2022, doi:10.3390/ph15111317_

Round 1

Reviewer 1 Report

Comparative studies of Palmatine with Metformin and 2

Glimepiryd on the Modulation of Insulin Dependent 3

Signalling Pathway In Vitro, In Vivo & Ex Vivo

The manuscript is interesting addressing the potential effects of palmatine, compared to metformin and glimepiryd, standard antidiabetic drugs, in cell culture and animal models of diabetes. Data is completed by in silico studies.

The are some aspects of the experiments that are not properly explained and there is a need for details in this respect. Also, authors are advised to re-check the quality of English language

The choice for the cell lines selected for the experiments should be justified.

The choice for expressing the concentrations of treatments used for animal studies should be justified. An exact dosage mg/kg body weight would be more appropriate.

For the cell culture experiment, the treatment should be more precisely described – were cells treated with only one conc for every substance (WHY???), how long were cells exposed to treatment, what were the positive and negative controls, what were the solvents used for stock preparation, etc. Did authors perform cell viability assays for the 3 tested substances?

Figure 1 is rather crowded – another mean instead of bbbb… to represent statistically significant differences should be found

Language corrections needed:

Metformin (biguanide) and glimepiryd (sulfonylurea) both drugs are derivatives of urea have been used as a first line drug for the treatment of T2DM???

Interestingly, Palmatine-treated cells/tissue/animas was able to reverse these effects.???

Derivatives of urea such as Metformin (biguanide) and glimepiryd (sulfonylurea), have been used as a first line drug for the treatment of T2DM.

The effect 414 of palmatine was stronger compared to metformin and glimepiryd that.???

\The potential 415 mechanism of action is through reduced expression of PKC-α with concomitant 416 increased tyrosine phosphorylation and thus activating IRS-1-PI3K-AKT2 cascade which 417 could ultimately enhance GLUT4 translocation.

 Based on the binding energies it can be concluded that palmatine have good affinity

Author Response

All reviewers queries have been addressed as attached. Thank you .

Reviewer 2 Report

I have read the manuscript carefully. An interesting study examining palmatine in vitro, in vivo, and ex vivo to obtain the molecular mechanism of its antidiabetic activity. Nonetheless, the manuscript suffers from poor writing; many typos and grammatical errors were found. I suggest authors to use professional proofreading service. Moreover, authors have to answer my concerns presented below.

Major

1.     Please briefly describe the study design in the Introduction. This includes the justifications for each protocol; i.e. the use of skeletal muscle tissues. Authors should also justify the use of molecular docking; authors may refer to the following study reporting the employment of molecular docking of natural compounds against α-glucosidase: Andalia et al. 2022 Karbala International Journal of Modern Science 8 (3), 330-338

2.     Please improve the Results description. Authors may calculate percentage or how many times one variable is higher or lower than another.

3.     In-silico results have NO DESCRIPTION, how so? Also, explain why proteins P3IK-alpha and -gamma were used. I found the descriptions in the discussion, kindly moved them to the results!

4.     I encourage authors to present their results in Figure 1 as table. Also, did you measure pre- and post-treatment level. Otherwise, normal control should be provided. I am wondering about the downregulation or upregulation terms if authors do not present the data before the STZ injection.

5.     Table 3. Palmatine has H-bond acceptor >10, hence violating Lapinski’s rule of five. Moreover, check again Line 136—13; why palmatine was assigned with (-) for the Ghose and Veber?

6.     Discussion for T2DM is too long. Moreover, since the study is about molecular mechanisms kindly provide the schematic diagram for the mechanisms.

7.     To make a clear discussion, authors are suggested to present the results obtained from their present study and then compared them with the results from published literatures.

8.     Why parameters such as blood glucose, glycated blood glucose, and AGE were not measured? Since an in-vivo study was carried out.

9.     Statistical analysis is unclear. The normality test should be carried out before tested with ANOVA.

10.  I suggest the results to be re-modified. Present the results based on in-vitro, in-vivo, and ex-vivo. I want to see the comparison the authors made. Also present the results in Table instead.

Minor

1.     Not sure if the journal employs structured abstract. Please confirm the guidelines.

2.     Abstract. Avoid the use of abbreviations “IRS- PI3K- IRS-1 -PKC -AKT2, and GLUT4”, otherwise defined.

3.     Line 23-24. Please confirm the accuracy of “both drugs are derivatives of urea”. Also, kindly correct the grammatical errors.

4.     Line 25. “Current study” consider “This present study”

5.     Line 27. “in vitro, in vivo & ex vivo” consider “in vitro, in vivo, and ex vivo”. Also, check the entire manuscript. Consider to replace ‘&’ with ‘and’.

6.     Line 33. qPCR should be defined first.

7.     Please be consistent with the writing of palmatine; use capital or simple letter?

8.     What (+) and (-) represent in Table 3? Elaborate in Table footnote.

9.     Line 177. Please complete “PIP3 (…)”

10.  Line 292. “CO2” please subscript ‘2’

11.  Line 288. “subculture in 2-3 days” consider “subcultured in 2-3 days”

Author Response

All Reviewers concerns has been addressed 

Round 2

Reviewer 1 Report

Authors responded to previous questions and improved the manuscript. It would be a good ideea to re-check the language quality. For example, authors state in a response "Derivatives of urea such as Metformin (biguanide) and glimepiryd (sulfonylurea), are wildly used as first line drug for the treatment of Type 2 Diabetes Mellitus." The word wildly should be replaced with widely or largely

Author Response

Responses uploaded 

Reviewer 2 Report

Some of the comments persist unanswered:

1.     The justification of the study design should answer the question ‘why?’ For instance, why molecular docking was used? Why the L6 skeletal muscle was used.

2.     Results 2.1 – 2.5. The results elaboration should be improved. Please state how much the increase/decrease occurred.

3.     Authors state that blood glucose, glycated Hb, and AGE have been reported separately. This is an indication of ‘salami slicing’. Kindly provide justification as to how the two publications answer different research question.

Author Response

Response uploaded 
